



# The terrestrial ice margin morphology in Kalaallit Nunaat (Greenland)

Jakob Steiner[1], Jakob Abermann[1], and Rainer Prinz[2]

[1]Institute of Geography and Regional Science, University of Graz
[2]Department of Atmospheric and Cryospheric Sciences, University of Innsbruck

**Correspondence:** Jakob Steiner (jff.steiner@gmail.com)

**Abstract.** The Greenland ice sheet (GrIS) and its peripheral glaciers and ice caps (PGIC) have received a lot of attention with respect to its marine-terminating, and considerably less for the remaining sections ending on land or in lakes. While the dominant part of ice mass imbalance is driven by calving at marine termini, a large part of the mass loss is caused by surface melt, leaving via those latter less studied margins. Relying on ice masks and a dataset for lake distribution we for the first time provide an assessment of the lengths of marine-, land- and lake-terminating margins across Greenland, showing that over a total length of 76154 km and 174425 km, for GrIS and PGIC respectively, 96.4 % (93.1 % and 97.8 %) of the margin is land-terminating, with the marine- and lake-terminating margin making up only 2.2 % (3.6 and 1.6 %) and 1.4 % (3.3 and 0.6 %). We also show that the ArcticDEM product is able to capture margin morphologies across large parts of the land-terminating margin, identifying 28.4 % as near-vertical features over shallow terrain, confirming earlier hypothesis of a large prevalence of these extremely steep features. 13.4 % are identified as steep (∼20-45°) and 17.3 % as shallow ramps (<20°). These data provide a basis to investigate the reason for surface morphology differences at terrestrial ice margins.

## 1 Introduction

While a large part of the ice margin of the Greenland Ice Sheet (GrIS) and peripheral glaciers and ice caps (PGIC) is land-terminating, this fraction has so far never been quantified and has received much less attention than the marine-terminating margin. Most of the ice discharge in Greenland originates from marine-terminating outlet glaciers (Mankoff et al., 2021; Mouginot et al., 2019; Shepherd et al., 2020), but approximately one half to two thirds of the total mass loss can be attributed to surface mass balance change, which predominantly leaves the ice as meltwater over the surface to proglacial rivers and streams (van den Broeke et al., 2016; Lewis and Smith, 2009; Mankoff et al., 2021; Shepherd et al., 2020). GrIS margins have experienced widespread thinning in recent decades (Hanna et al., 2020; Kjeldsen et al., 2015; Shepherd et al., 2020) and marine-terminating tongues generally thin at higher rates than land-terminating ones (Sole et al., 2008) and have seen widespread retreat (Goliber et al., 2022; Howat and Eddy, 2011). 75 % of land-terminating drainage contributes to surface flow directly, while the rest drains into marginal lakes (Lewis and Smith, 2009).





Mouginot et al. (2019) provide an assessment of GrIS mass loss between 1972 and 2018, partitioning the ice sheet into 260
basins and regions. 217 basins have a marine-terminating outlet and 43 are completely land-terminating, with no (or negli-
gible) mass loss due to dynamic discharge at the terminus. Mass loss and change of dynamics were found to be temporally
variable, with periods of near-balance before the 1990s. Areas that are found to be stable in South-West Greenland, coincide
with parts of the GrIS where previously an advancing margin was identified (Knight et al., 2000; Weidick, 1991, 1994). As
marine-terminating glaciers retreat, a large part of their terminus becomes land-terminating, suggesting a potential future in-
crease of land-terminating drainage (Mouginot et al., 2019).

Several studies have previously focused on the Holocene fluctuations of the land-terminating ice margin or on its hydrolog-
ical impacts (Carrivick et al., 2018; Davison et al., 2019; Koziol and Arnold, 2018; Lesnek and Briner, 2018; Sole et al., 2008;
Tedstone et al., 2015; Weidick, 1968). Some field studies show stable or advancing ice margins of land-terminating glaciers
and ice caps in North Greenland throughout parts of the $20^{th}$ and $21^{st}$ century (Abermann et al., 2020; Davies and Krinsley,
1962; Dawes and As, 2010; Farnsworth et al., 2018; Goldthwait, 1971). At the Nunatarssuaq Ice Cap, a net ice margin ad-
vance on a centennial scale was postulated based on dated organic material (Goldthwait, 1961) and phases of advance, retreat
and re-advance during the past six decades were recently quantified (Abermann et al., 2020), coinciding with the presence
of vertical ice cliffs. Stability (or even advance) was found to go in line with thinning of the ice margin. In West Greenland
similar advances have been observed but generally associated with a thickening due to presumed increased precipitation with
increasing temperatures and moisture (Dawes and As, 2010; Tatenhove et al., 1995; Weidick, 1991).

Davison et al. (2019) and Koziol and Arnold (2018) note, that the understanding of ice dynamics on the terrestrial margin
(including land- and lake-terminating sections when looking at larger domains) is still confounded by an inadequate under-
standing of feedbacks between runoff and ice dynamics, but predict a slowdown of the margin with increasing melt. This
has previously been observed, suggesting that the land-terminating margin is more resilient to dynamic impacts of enhanced
melt (Tedstone et al., 2015). Other studies on the ice margin in Central-West Greenland emphasize an important link between
understanding margin dynamics, lake evolution and sediment evacuation from the GrIS (Carrivick et al., 2018; Knight et al.,
2000). Beyond the interest in contemporary ice sheet health, the terrestrial margin in North Greenland has also received some
attention in studies investigating paleo-climate, with a link between margin characteristics and recent Holocene climatic change
(Farnsworth et al., 2018; Lesnek and Briner, 2018; Osterberg et al., 2015; MacGregor et al., 2020; Reeh et al., 1987, 2002).

Studies specifically describing the very steep land-terminating margin sections have been conducted in the Antarctic Dry
Valleys (Levy et al., 2013) and North Greenland (Abermann et al., 2020). Ice cliffs along the margin of glaciers and ice sheets
are a widespread but rarely investigated feature (Weidick, 1994), that could potentially shed some light on the role of atmo-
spheric drivers or ice dynamics along the margin. Goldthwait (1960) estimates that approximately 45 % of the ice sheet in
Northwest Greenland terminates as cliffs on land. A small number of studies in Antarctica have investigated land-terminating
ice cliffs on the glacier margin in the Dry Valleys (Fountain et al., 2004; Levy et al., 2013; Swanger et al., 2017). A recent



advance of the margin, with pronounced vertical cliffs has been associated to recent warming periods, with a corresponding
thinning of the ice mass. They can significantly contribute to ablation, where the cliff makes up only 2% of the ablation area,
but constitutes for up to 20% of total ablation (Lewis et al., 1999).

The most comprehensive analysis of land-terminating margin morphologies to date was accomplished in West Greenland
(Nobles, 1961; Weidick, 1963, 1968). Investigating five localities along the Western coast between Thule in the North and
Iviangerquitit Tasiat in the South, Weidick (1963) finds ice cliffs ($vertical$ or $near-vertical$ following Nobles (1961), who
investigated different types of ice margin in the Thule region), steep ($steep\ ramps$, $20-45°$) and gently sloped margins ($gentle$
$ramps$, $<5°$). Steep ramps have been described as an intermediary stage between a decaying ice cliff into a gentle ramp (Nobles,
1961), while the evolution of vertical faces was explained as the result from overriding of stagnant by active ice (Goldthwait,
1961; Rausch, 1958), solar radiation (Chamberlin, 1895) or wind erosion Bishop (1957). To date no universal explanation
exists for morphological developments of steep to vertical ice margins (Steiner et al., 2022). Following initial observations
(Weidick, 1963) a number of characteristics can be noted that seem to apply for many steep sections along the margin in
Greenland, sorted by order of prevalence across sites: (a) ice cliffs are generally 10 to 40 m high; (b) steep sections appear
on the inland margin proper rather than on glacier lobes; (c) terrain adjacent to steep margin is defined by presence of shear
moraines and undulating ground moraines with round boulders rather than marginal or terminal moraines; (d) this topography
may favor strong local winds contributing to morphology, potential erosional effect of snow; (e) limited recession during the
first half of the $20^{th}$ century, at times even advance; (f) steep ramps are occasionally bordered by melt water streams or lakes;
(g) ice cliffs potentially form at the maximum limit of glaciation in a specific region; (h) mean ice temperatures are slightly
above mean annual air temperatures.

While a number of topics relating to the land-terminating ice margin in Greenland are of interest to understand past, present
and future changes of the GrIS itself, there is yet no comprehensive assessment of the margin's location and morphology that
would allow for a discussion of Weidick's observations on a larger scale. As Weidick (1963) notes, morphologies vary between
locations and therefore suggests to analyse larger parts of the margin. With recent delineations of the complete margin (Citterio
and Ahlstrøm, 2013; Rastner et al., 2012) as well as high resolution data of elevation (Morlighem et al., 2017; Porter et al.,
2023) such an investigation becomes now possible. This study therefore attempts to quantify how much of the GrIS and PGIC
in Greenland is terrestrial (including land- and lake-terminating parts), where steep and gentler sections of the land-terminating
margin are located and how variable slope is along the margin (Figure 1), to provide a baseline for future studies to further
investigate the characteristics of the margin as described in Weidick (1963).





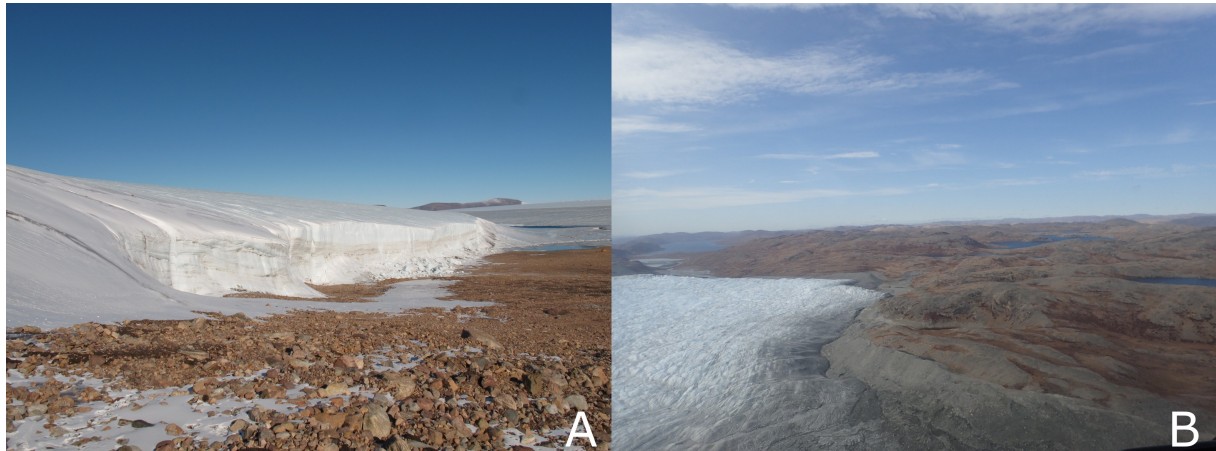

**Figure 1.** (A) Transition from a steep ramp to a vertical ice cliff back to a steep ramp (what we define as 20 to 45°) within a few 100s of meters at the Nunatarssuaq Ice Cap. The surrounding terrain is generally flat. Photo: Authors, 2017. (B) The ice margin at Russel Glacier, typical for an outlet glacier with reworked periglacial terrain, with predominately shallow to steep ramp margin features. Photo: Authors, 2019.

## 2 Data

A number of large spatial datasets have become available for Greenland in recent years that allow for an analysis of the ice margin, following openly available resources.

The outline of the GrIS (Citterio and Ahlstrøm, 2013) was complemented with outlines of PGIC (Rastner et al., 2012), both corresponding to the year 2000 (1999-2002), which aligns to the time stamps of a Landsat mosaic available to check margin
quality. For ease of processing and analysis, the margin is separated into the basins of the GrIS (Mouginot et al., 2019). An overview over these basins and the regions they are associated to is provided in Figure S13. To distinguish between marine- and land-terminating margin we relied on the BedMachine Greenland version 5 (Morlighem et al., 2017). To identify ice margin potentially in contact with lakes we intersected with the existing lake inventory (How et al., 2021), which was manually corrected to align with the 2000 time stamp. All input data reprocessed for this study is available in the corresponding data
repository (Steiner et al., 2025).

ArcticDEM provided surface elevation data, specifically the Mosaics version 4.1, which are available at 2 m resolution (Porter et al., 2023). Mosaics are produced from image composites between 2012 and 2022, implying a potential mismatch with margin products and imagery. As it is not feasible to produce new margin outlines corresponding to the DEM time
stamps, nor are DEMs available around the year 2000, we overcome this challenge by working with a 100 m buffer within the margins (see Methods). Sections where the margin outline does not correspond to the actual margin are flagged and not used



to extract margin morphologies. A total of 448 ArcticDEM tiles (see Figure S1) are used and converted into slope maps using the terra::terrain() function (Hijmans, 2025) in R with 8 neighbor cells. Slope values below 5° are ignored for identifying the eventual margin morphology to avoid including too much flat terrain adjacent to the actual margin, either on land or ice. The
Pléiades stereo-pairs, with processed DEM, used in this study for two validation sites in North and East Greenland, respectively, were provided by the Pléiades Glacier Observatory initiative of the French Space Agency (CNES, Berthier et al. (2024)).

## 3 Methods

The process to delineate the ice margin and extract the slope from the terrestrial margin is visualized in Figure 2. The margin is separated into 260 basins, of which 40 have no or negligible margin that was merged into other subbasins, 204 are connected
to the GrIS, while the remaining 16 are associated to PGIC (step A, Figure 2). A buffer of 100 m is placed on the inside of the mask to avoid including off-glacier terrain (step B). Instances where the margin mask is smaller than the actual ice extent are found to be rare upon visual inspection. Python scripts to obtain the buffer are documented in the Supplementary Material. During this step, the margin is intersected with polygons of lakes, ocean and any directly adjacent ice bodies, like PGICs connected to the GrIS. To determine the actual terrestrial margin, the BedMachine DEM (Morlighem et al., 2017) is used at its
resolution (150 m). Any part of the margin located above 10 m a.s.l. is considered terrestrial, and the rest marine-terminating (step C). This margin allows for a conservative estimate of the terrestrial margin, rather than overestimating it, in cases where the ice mask is not accurately placed. The product is compared manually against the Landsat orthomosaic for validation, and any erroneous classification was manually corrected in the vicinity of marine outlets. Parts of the margin that intersect with a lake are considered lake-terminating. Due to the later time stamp of the lake inventory (2017, How et al., 2021), all marginal ice
contact lakes were checked with the imagery, missing lakes added (n=302) and lakes not present in 2000 removed (n=1151). The updated lake inventory is available with the data repository (see Section 6). Note that we did not remap the margin and hence, due to their ephemeral nature, changing rapidly sometimes even within one year, the absolute lengths of marine- and lake-terminating margins provided here should be interpreted with caution. Margin outlines were only corrected where erroneous outlines appeared on ice (obvious processing errors rather than misclassifications), and margins that were duplicated
between the GrIS and PGIC mask (predominately in East Greenland) were removed from one of the two. Lengths of terminus types are calculated for each subbasin (step D). We include margins along nunataks (i.e. closed polygons within the respective margin), but report statistics with, and without considering these sections for the GrIS). For the final step of slope extraction, only terrestrial margin buffers are retained, along which 1 km grid cells are placed (step E). Slope maps are produced for each ArcticDEM tile and used to then extract slope values for the buffer within each grid cell (step F), stored as individual text files.

Because the ice margin is not derived from the ArcticDEM, three issues need clarification to allow validity of the presented method of investigating the ice margin morphology. The first question is whether the existing margin masks represent the actual margin. This issue poses a general challenge in glacier ice delineation, faced by any larger dataset covering more than a few individual glaciers, and can not be comprehensively assessed here. However, visual inspection of the margin on the mosaic







**Figure 2.** Workflow to extract margin morphologies across the complete ice margin (example basin No. 70). (A) Basin outlines and ice masks are intersected to produce margins for all subbasins. (B) A 100 m buffer is placed along each margin section. (C) The margin is intersected with the Bedmachine DEM to identify marine-terminating, and lake polygons to identify lake-terminating margin. Margin intersecting with other glacier ice (e.g. ice sheet with a peripheral glacier) is also stored separately. (D) Relative margin lengths for each subbasin are saved separately for each subbasin, and only land-terminating margin is retained for final steps. (D) A 1 km grid is placed over the terrestrial margin polygon, to extract data from underlying ArcticDEM tiles. (E) Slope values are extracted for each grid cell from the slope maps, resulting in slope distributions (inset) for each grid cell.





produced from Landsat images from around the year 2000, indicates that the GrIS land-terminating margin is captured well in most cases, and only diverges visibly in regions with very winding margin sections or for cases where the shallow margin is loaded with debris. This being generally regions with very shallow slopes, we believe that it does not significantly impact our results. The margins of PGICs are of poorer quality, with more misclassifications of (perennial) snow fields as ice. To avoid the inclusion of snow fields and very small glaciers, where margin morphology is arguably of no more interest to understand

glacier dynamics, we have removed all separate ice bodies with an area smaller than $5\,\mathrm{km}^2$, amounting to a total of $12429\,\mathrm{km}^2$ or, $3.5\,\%$ of the total area of all PGICs. Furthermore, any grid cell that contained margins that was visually obviously misplaced, either due to misclassification or a consistent offset between margin outline and actual ice margin was marked manually and morphology was not considered in these cases.

The second question is whether a $100\,\mathrm{m}$ buffer inside the margin is sufficient for slope detection analysis. As the DEM is obtained from imagery after 2000, with a horizontal accuracy of $4\,\mathrm{m}$ and with a predominately retreating margin, setting such a wide buffer inside the margin allows us to capture the margin also in the DEM. Since, however, we are predominately interested in the slope of the very front of ice sheet or glaciers, it is important applying a buffer size reducing the number of pixels of flat ice surfaces. Visual inspection across the GrIS suggests that ice masks rarely intersect the actual ice, hence excluding any area

outside the margin is warranted and further reduces misclassifying pixels. The time stamp of the ArcticDEM, produced from imagery between 2008 and 2022, differs by more than a decade from the time stamp of the margin. Margin recession rates of 4 to $13\,\mathrm{m}\,\mathrm{yr}^{-1}$ within this period would hence be still covered within a $100\,\mathrm{m}$ buffer, which seems reasonable given approximate rates of ca. 10 - 30 $\mathrm{m}\,\mathrm{yr}^{-1}$ in very active regions (Mernild et al., 2012).

Finally, we need to ascertain that the quality of the ArcticDEM suffices to capture steep margin sections. For two sites in Greenland, the Nunatarssuaq area (Abermann et al., 2020) and the Mittivakkat Glacier, high resolution Pléiades imagery is available (see Supplementary Material for details) allowing for an analysis of the ability of ArcticDEM to capture margin morphologies. The aim of this comparison was not whether the margin outline matches with the DEM, but rather if the resolution of the ArcticDEM is able to capture relatively steep morphologies. To do so we mapped steep sections in both regions, placed

a $100\,\mathrm{m}$ buffer around the delineation and compared the respective slope values.

To identify near-vertical margin sections, steep and shallow ramps from the DEM, we extracted cumulative distribution functions of slope values for well known margin sections across field sites in Nunatarssuaq (Abermann et al., 2020), Inglefield Land (Kjær et al., 2018), Freya Glacier (Hynek et al., 2023) and Qaamarujup Sermia (Abermann et al., 2023). The slope

distributions for all cells are then evaluated with the DTS test, a version of the Wasserstein metric, which essentially provides a least squares of the slope distributions (1) (Dowd, 2020).

$$DTS_i = \int\limits_0^{90} \frac{|\hat{F}_i(\beta) - \hat{E}_i(\beta)|}{\hat{D}_i(\beta)(1 - \hat{D}_i(\beta))} d\beta \qquad (1)$$





The theoretical slope distribution $\hat{F}(\beta)$ that fits the sample $\hat{E}(\beta)$ distribution for grid cell $i$ best determines whether the margin section is most likely near-vertical, steep or shallow. $\hat{D}(\beta)$ is the variance of the sample. This method avoids misin-

terpreating periglacial terrain that has similarly steep slopes as the ice margin, and which would hence have similar average slope values. Although this approach does not allow us to identify the type of margin for each pixel of the margin, it identifies a dominant margin type for each 1 km grid cell. We do not extract slope distributions for parts of the margin earlier identified as erroneous, nor for grid cells where the bed topography is on average steeper than 30°. This avoids the inclusion of margins especially prevalent around mountain glaciers, where the distinction between steep terrain and ice margin is difficult and

temporal changes in ice margin arguably faster than elsewhere, hence making our assumption of the suitability of ArcticDEM mosaics for this method questionable.

## 4 Results

### 4.1 Absolute and relative margin lengths

The total length of the GrIS margin is 76154 km (containing a total area of 165945 km$^2$), 93.1 % of which is land-terminating,

3.6 % marine-terminating and 3.3 % lake-terminating (Table 1, Figure 3). The land-terminating fraction is generally higher in East (between 93.2 and 97.9 %) than in West Greenland (85.1 % to 90.8 %), but varies widely between basins (Table S2), ranging all the way from 24.1 % (basin 50 in NO, with marine-terminating margin making up 71.3 %) to 100 % in 5 basins, all located in NW Greenland. Marine-terminating fractions range from 0 % (in 21 basins) to 71.3 % (basin 50), while lake-terminating fractions range from 0 % (72 basins) to 21.6 % (basin 69, SW). In SW the lake-terminating fraction of the margin

is larger than the marine-terminating fraction. 4160 km of the ice sheet is connected to peripheral glaciers and ice caps. These segments are not included in any of the following statistics.

The margin around PGIC at 174425 km (12429 km$^2$) is nearly twice as long as the GrIS margin, with 97.8 % land-terminating, 1.6 % marine- and only 0.6 % lake-terminating (Table 1, Figure 3). There is no difference in relative lengths between the eastern

and the western regions but PGIC in NW and SE have especially long marine-terminating margins (4.6 and 3.9 %, respectively), while in SW lake-terminating are longer than marine-terminating margins (1.4 and 0.5 %, respectively). Depending on the existence or absence of marine-terminating outlet glaciers and ice streams, the relative fraction, however, also varies widely within regions (Figure 3, Table S2).

### 4.2 Ability to capture margin morphology

The ability to capture the morphology of the margin with the methods used is defined by (a) the accuracy of the margin outline, (b) the adequacy of the ArcticDEM resolution and quality to distinguish between steep and shallow sections and (c) the created buffer's ability to actually capture the ice margin rather than any adjacent terrain off or stretches on the ice. The first aspect we can not definitely quantify here, but it becomes apparent from visual inspection across Greenland, that especially along shal-





**Table 1.** Lengths of margins for all regions on the GrIS as well as the PGICs, including the relative fraction of the margin ending on land (*land − terminating*), in the ocean (*marine − terminating*) and in a lake (*lake − terminating*). In brackets are values when ignoring margins along nunataks (i.e. closed polygons within the respective margin), only calculated for the GrIS. The number in italics in brackets denotes the fraction of the land-terminating margin that is considered of insufficient quality to extract margin morphologies. See Table S2 in the Supplementary Material for details on each individual subbasin.

| Region | total margin [km] | land-terminating [%] | marine-terminating [%] | lake-terminating [%] |
|---|---|---|---|---|
| GrIS CW | 2315 | 85.1 (85.6, *19.3*) | 8.8 | 6.1 |
| GrIS NW | 7470 | 85.4 (83.9, *15.9*) | 12.2 | 2.4 |
| GrIS NO | 5644 | 85.5 (80.6, *20.4*) | 7.7 | 6.8 |
| GrIS NE | 19340 | 93.2 (92.8, *5.7*) | 1.6 | 5.2 |
| GrIS CE | 16381 | 97.9 (97.1, *2.4*) | 1.3 | 0.8 |
| GrIS SE | 17219 | 96.4 (89.2, *7.2*) | 3.4 | 0.3 |
| GrIS SW | 7785 | 90.8 (90.4, *13.4*) | 1.3 | 7.9 |
| TOTAL GrIS | 76154 | 93.1 (92.9, *8.4*) | 3.6 | 3.3 |
| PGIC CW | 14068 | 99.4 (-, *1.9*) | 0.4 | 0.3 |
| PGIC NW | 5639 | 93.9 (-, *23.5*) | 4.6 | 1.4 |
| PGIC NO | 20132 | 97.2 (-, *12.4*) | 1.5 | 1.3 |
| PGIC NE | 49887 | 98.3 (-, *3.4*) | 1.1 | 0.6 |
| PGIC CE | 47810 | 98.5 (-, *1.6*) | 1.4 | 0.2 |
| PGIC SE | 25267 | 96.0 (-, *2.9*) | 3.9 | 0.1 |
| PGIC SW | 11624 | 98.1 (-, *9.5*) | 0.5 | 1.4 |
| TOTAL PGIC | 174425 | 97.8 (-, *4.9*) | 1.6 | 0.6 |
| TOTAL | 250580 | 96.4 (-, *6.0*) | 2.2 | 1.4 |

low ramps of the GrIS in South Greenland as well as at the margin of a number of PGICs (Figure 4), there are non-negligible
sections of the ice masks that do not match with the actual ice margin visible from imagery. After removing margin sections
that are nonsensical (Figure 4A), we manually identified those grid cells that cover visibly erroneous margin. 8.4 % of the GrIS
and 4.9 % of the PGIC margins are potentially erroneous (Table 1) and are not considered to investigate the morphology further.

The second requirement we ascertain by a comparison to Pléiades imagery with the same 2 m resolution as ArcticDEM.
Figure 5 and Table 2 show slopes in both products in two field sites (see Supplementary Material for details on data). While
neither timing nor native resolution of the DEMs are the same, slope maps match well, with Spearman correlation higher than
0.75. Both products retain maximum slope values beyond 60° in known vertical sections of the margin, suggesting that the





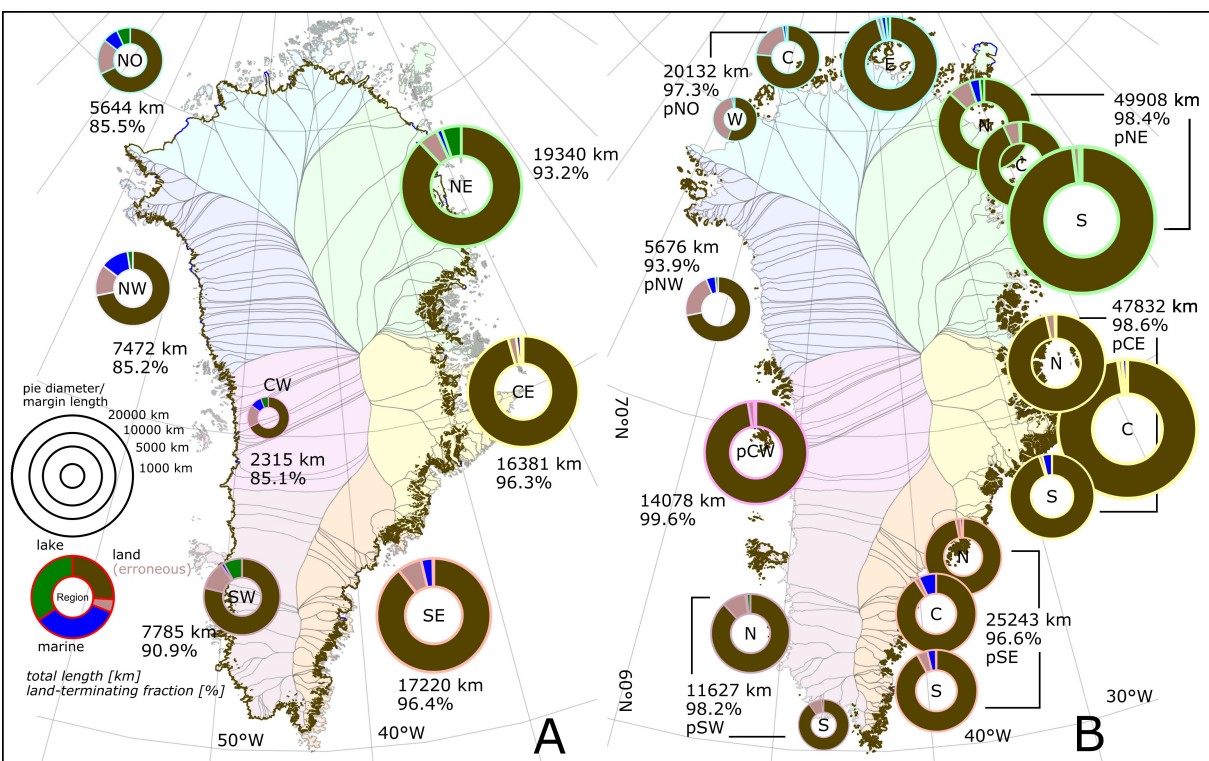

**Figure 3.** Margin lengths and respective land-, marine- and lake-terminating fractions for the GrIS (A) and PGIC (B). The numbers adjacent to the pie plots refer to total length and relative fraction of the land-terminating margin. Note that for PGIC the lengths are shown separately for subsections of the pNO, pNE, pCE, pSE and pSW regions, corresponding to data shown in Table S3, to avoid excessive size of pies (for pNE, pCE, pSE) and due to the disconnected nature of ice masses (pNO, pSW). Values of individual subbasins for the GrIS regions for PGICs are shown in Table S3.

**Table 2.** Statistical comparison between slope values extracted from a buffer around the margin of two focus areas (see also insets in Figure 5). n is the total number of pixels that are overlapping in both products.

| study site | n | median slope [°] (ArcticDEM/Pléiades) | median difference [°] | bias [°] | spearman r [°] |
|---|---|---|---|---|---|
| Nunatarssuaq | 1073256 | 12.64 / 12.70 | 0.23 | 0.11 | 0.87 |
| Mittivakkat | 1064084 | 12.65 / 13.27 | 0.48 | 1.18 | 0.78 |

ArcticDEM is able to catch (near-)vertical sections of the ice margin. This supports the hypothesis that the ArcticDEM is suitable to derive margin morphologies at a larger scale.






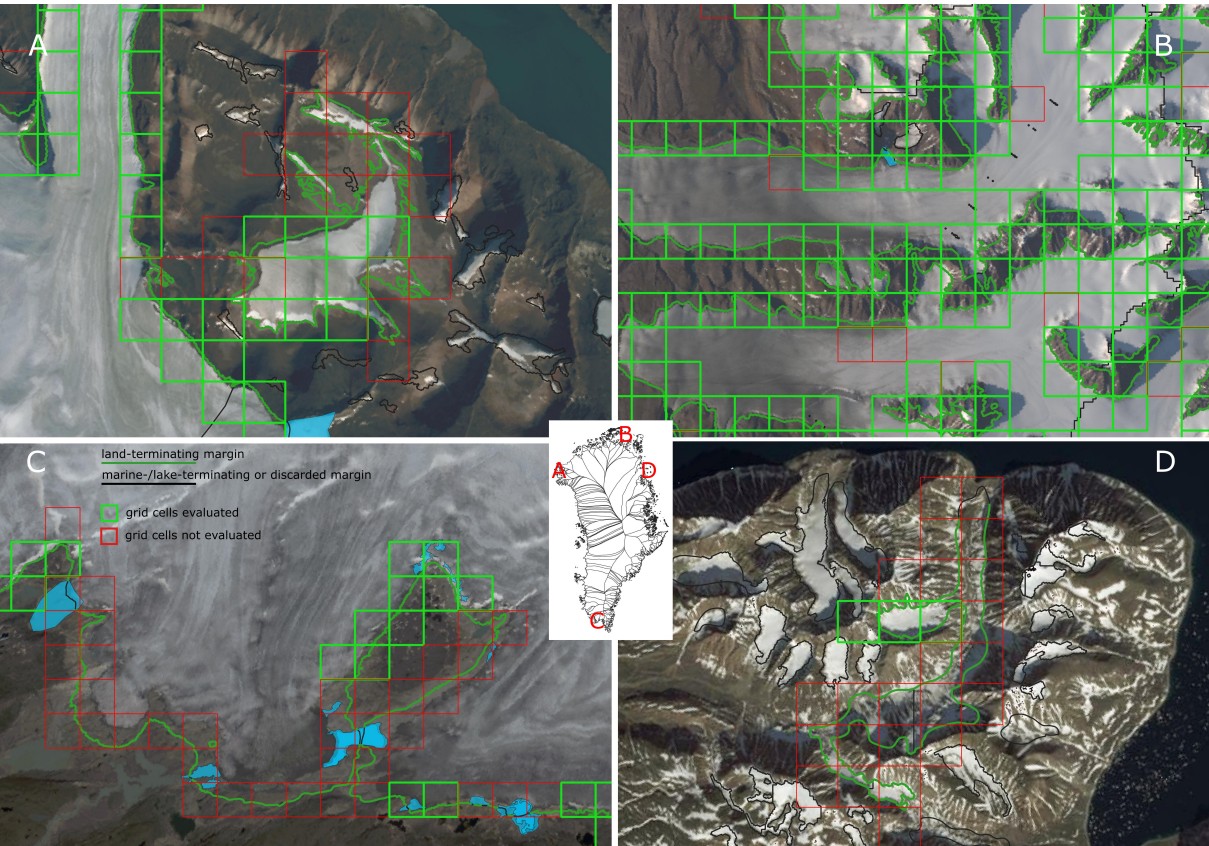

**Figure 4.** Examples of incorrect terrestrial margin outlines. Green is the already extracted terrestrial margin, black is margin that was not considered in any of the analysis (e.g. discarded obvious artefacts as the black dots in subpanel A, margins of individual disconnected glaciers that fall below the size threshold as in subpanel C or any part of ice masks that connects GrIS directly to PGIC). Grey grid cells are discarded in the analysis of margin morphology, after manual identification of ice mask mismatch with actual margin in imagery. (A) Erroneous margin pixels on an outlet glacier. Two small glacierettes in the West of the extent were discarded due to their size. (B) Margin in North-West Greenland (PGIC), where snowfields or ice debris from dry calving were included in the ice mask in many locations. (C) Margin in North-East Greenland (PGIC) with potentially inconsistent mapping methodology. The southern part shows a meticulous mapping at pixel scale, whereas the northern part is highly generalized. (D) Margin in South Greenland (GrIS) with a systematic offset, possibly pointing to a temporal difference between image and margin detection.

The final criterion to evaluate whether the available data can potentially capture margin morphologies is whether the margin buffer is able to capture the actual ice margin. To test this, we placed transects across 41 random locations of the buffer (Figure 6, see Supplementary material for locations and process). Across the margin there are two types of clear misclassifications, when the actual margin is missed by the buffer, which makes up 7 % of the samples taken (cases A and B in Figure 6). This corresponds well to the overall fraction of erroneous margin identified for all of Greenland (6.0 %, Table 1). Another 10 % are cases where more than half of the margin buffer based on the ice masks captures periglacial terrain with different slope





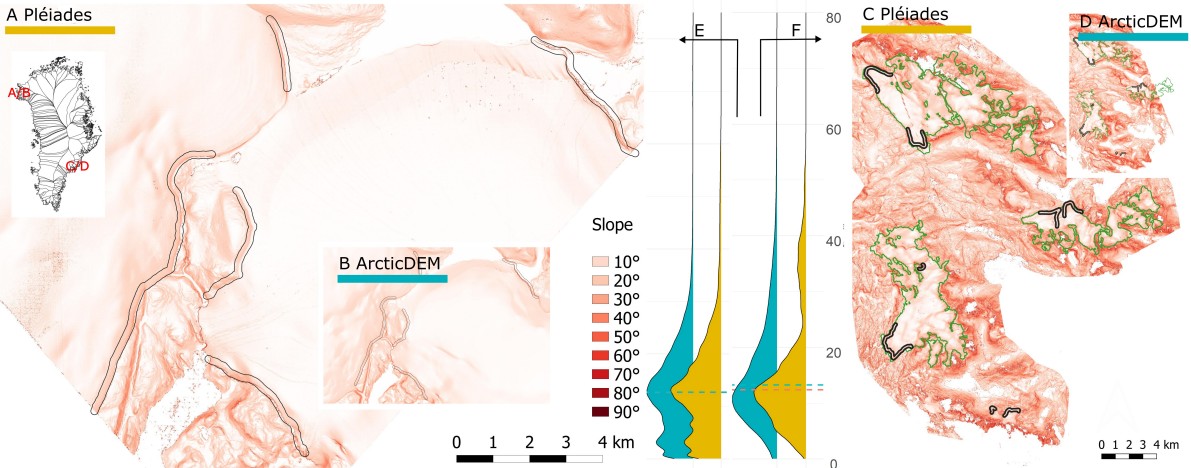

**Figure 5.** Comparison of slopes at the Nunatarssuaq ice cap (A,B) and Mittivakkat glacier area (C,D; see inset for location in Greenland) derived from the Pléiades and ArcticDEM, respectively. The slope distributions (E, F) show the distribution of slope values within the buffer around the margin, shown in the map panels.

than the ice (cases C and D). Cases E and F show typical transects at shallow margins. While here ice-free terrain is often included in the buffer, it does not lead to misclassifications if the ice-free terrain has similar steepness. In many regions the buffer may only sample a limited section of the ice margin, but it represents the dominant slope (G, H). To not bias samples towards shallow proglacial terrain or shallow sections of the ice behind the margin, slopes below 5° are discarded from the morphological analysis. Similarly, cases I and J are typical steep sections, where always some shallower terrain (on or off-ice) will be included.

### 4.3 Identifying spatial margin morphology

It is apparent from above that calculating the average slope across a certain section would not return a useful metric for describing margin morphology. It would be impossible to tell when the data in case of steep margins are biased towards shallow parts of the proglacial terrain or the ice surface included in the buffer or when shallow segments are misclassified as steep due to the inclusion of steep ice-free terrain adjacent to the ice. We therefore establish exemplary cumulative distribution functions (CDF) for four types of margin from a total of 40 1 km grids (with a total of 741 225 2 m cells), namely near-vertical margin on shallow terrain (21 %, example $C_b$ in Figure 7), near-vertical margin on steep terrain (5 %, example D in Figure 7), steep ramps (26 %, $C_a$) and shallow ramps, both around ice caps (31 %, A) and a mountain glacier (17 %, B). We use the term 'near-vertical' to include margin sections that are vertical as well as very steep (roughly >45°), and 'shallow' versus the term 'gentle' used previously only for ramps <5° (Nobles, 1961). These distributions were obtained from grid cells where margins are well known from field studies (Figure 7). Shallow margins are predominately below 20°. However, ramps located along margins with relatively flat proglacial terrain (as in Inglefield, Figure 7), still have a distinctly different slope distribution than those with





**Figure 6.** Example profiles intersecting perpendicular to the margin, the line representing the surface. The red part marks the surface that intersects the margin buffer, the blue shade indicates ice cover and blue vertical lines the position of the ice margin. Cases A and B are considered misclassifications, cases C and D represent considerable offsets with large parts of ice-free terrain captured by the buffer, while all others capture the actual ice margin well.

adjacent mountain slopes. Slopes on steep ramps are clustered below 20° and upper quantiles are similar to shallow ramps, but the median is distinctly higher. Near-vertical sections on shallow terrain can be distinguished by higher fraction in upper quan-



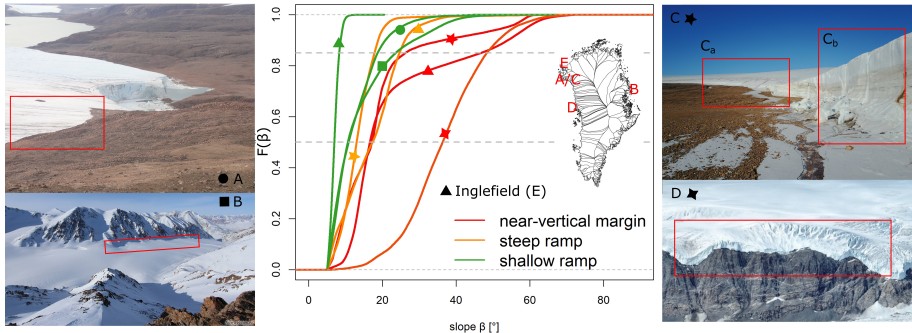

**Figure 7.** Cumulative distribution functions for selected margin sections, identifying near-vertical cliffs (red), steep (orange) and shallow ramps (green). Symbols refer to regions where margin types were sampled, shown in the four photos. Horizontal dashed lines are the $50^{th}$ and $85^{th}$ quantile. Red squares in the photos denote the approximate margin areas selected. Locations A and C are at the Nunatarssuaq ice cap (Abermann et al., 2020), B is Freya Glacier, and D is located in West Greenland (Abermann et al., 2023, photo: Fauland/Wally). Locations in Inglefield Land are documented in Kjær et al. (2018) and Esenther et al. (2023).

tiles, while those over steep terrain - common around outlet and mountain glaciers but often less distinctly vertical than their counterparts on shallow terrain - have a higher median, which can be largely explained by the slope of the terrain rather than the ice margin itself. We consider this specific type of near-vertical margin over steep terrain separately for two reasons. It becomes

difficult to identify whether margin in these regions is predominately steep because of ice morphology or underlying terrain, and sections may be vertical ice falls while others may be steep ramps. These parts generally also do not include vertical ice cliffs that motivated previous investigations into the terrestrial margin Goldthwait (1960); Weidick (1963); Steiner et al. (2022).

A detailed presentation of grid cells used for the CDFs used for classification is provided in the Supplementary Material.

The CDFs have a median of 8° for shallow ramps, 14° for steep ramps and 17°/36° for near-vertical margin on shallow and steep terrain, respectively. The small difference between median values illustrates how a clear distinction would become difficult, using those alone. The $85^{th}$ quantile however (18°, 22°) corresponds to the rough definition of shallow ramps (<20°) and steep ramps (20-45°) by Nobles (1961). Figure 8 shows examples of the margin with the predominant margin morphologies identified. The algorithm is able to capture relatively swift transitions between shallow ramps, to steep cliff sections across a

terminus (Figure 8A). On the larger scale, sections well known as continuous cliffs for example along the Hiawatha crater, eventually transitioning to steep ramps and much more shallow ramps in Inglefield Land are captured by this approach (Figure 8B).

Across Greenland we find that along the land-terminating margin of the GrIS 24.8 % are near-vertical cliffs on shallow terrain

(another 16.1 % over steep terrain), 9.5 % steep and 20.2 % shallow ramps (while the remaining 29.0 % were not evaluated, due to erroneous margin masks or bed topography steeper than 30°, Table 3). There are however large differences between regions, with one third of the margin in West and North Greenland ending in shallow ramps, but only about 10 % in Central



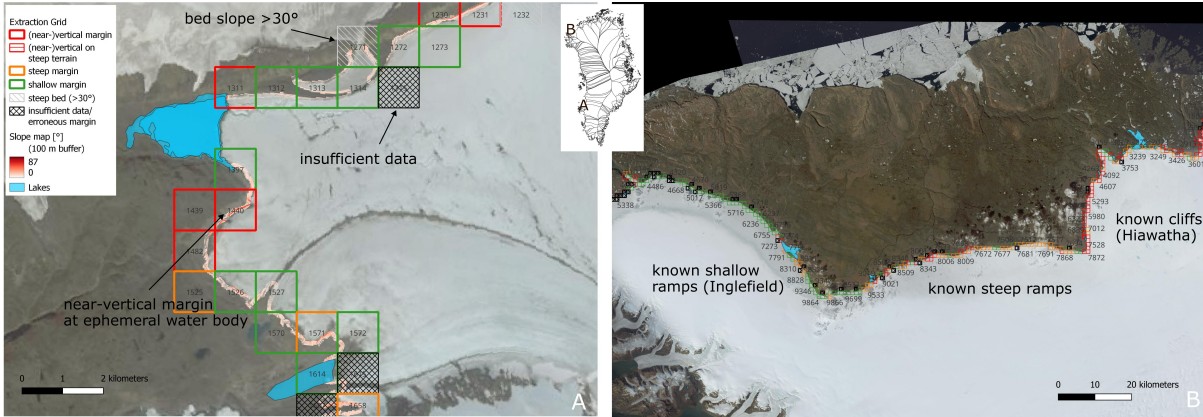

**Figure 8.** Examples of identified predominant margin morphologies in South-Western (A, catchment 4) and North-Western Greenland (B, catchment 176). Numbers in grid cells refer to the sequential numbers of cells under which the slope data is stored.

and South East Greenland, where instead near-vertical margins on steep terrain are more common. Individual basins have up to 50 % near-vertical margins on shallow terrain, while others have more than 70 % shallow margins. Shallow ramps are

especially prevalent across the northern margin and in South-West Greenland and largely absent in the more complex Eastern basins (Figure 9A). Near-vertical margins are also prevalent on PGICs in East Greenland (Table 3, Figure 9B), while more rare in North Greenland where shallow ramps are more common. Across all regions 19.2 % of the land-terminating margin morphology was not assessed due to the steep bed topography (13.2 %) or erroneous ice masks (6 %, Table 1 and 3). In 16 of the 220 basins less than 10 % and in 3 basins more than 90 % of the land-terminating margin was not classified.

**5    Discussion**

Locations and extents of marine termini in Greenland have been mapped in great detail for peripheral glaciers (Kochtitzky and Copland, 2022) and the ice sheet (Greene et al., 2024) and for both cases (Carrivick et al., 2022), all with different approaches and data. Kochtitzky and Copland (2022), manually measured flux gates across all outlets between 2000 and 2020 and find a total length of 697 km, Carrivick et al. (2022) mapped 615 km, slightly less than one third of the 2549.2 km mapped in our study

(Table 1). Carrivick et al. (2022) find 1698 km for the GrIS, roughly two thirds of the 2738.5 km mapped with our approach. These large differences can be explained by the earlier studies focusing on flux gates, while we consider any part of the ice mask located where the DEM is <10 m a.s.l. as marine-terminating, and subsequently also include lateral sections along fjords or convoluted margin polygons at the marine terminus. While our margin length should therefore not be used to estimate ice flux, these much larger margin segments describe the locations where the ice is in potential dynamic contact with the ocean.

Lake margins mapped by Carrivick et al. (2022) for the GrIS are 3176 km (compared to 2738.5 km mapped in our study) and for the PGIC 795 km (952.8 km), respectively. Although the original lake dataset used is the same (How et al., 2021), it was corrected for available imagery for 2000 in our study and the margin used in Carrivick et al. (2022) is coarser and works with





**Table 3.** Fractions of the terrestrial margin respectively classified as 'near-vertical' (in brackets values for near-vertical sections on steep terrain, see Figure 7D), 'steep' or 'shallow' ramps. NA refers to margin sections where the morphology was not investigated, either due to erroneous margin (see also Table 1) or steep bed topography.

| Region | land-terminating margin [km] | near-vertical (steep terrain) [%] | steep ramp [%] | shallow ramp [%] | NA [%] |
|---|---|---|---|---|---|
| GrIS CW | 1970 | 17.7 (4.5) | 11.2 | 31.2 | 35.4 |
| GrIS NW | 6382 | 27.7 (4.2) | 11.6 | 25.0 | 31.1 |
| GrIS NO | 4825 | 26.3 (2.3) | 11.8 | 32.9 | 26.8 |
| GrIS NE | 18030 | 26.6 (9.9) | 12.9 | 28.2 | 22.4 |
| GrIS CE | 16044 | 25.6 (30.2) | 5.9 | 10.0 | 28.3 |
| GrIS SE | 16591 | 22.0 (24.7) | 5.6 | 7.3 | 40.5 |
| GrIS SW | 7072 | 22.8 (3.2) | 14.1 | 37.0 | 22.9 |
| TOTAL GrIS | 70913 | 24.8 (16.1) | 9.5 | 20.2 | 29.4 |
| PGIC CW | 13977 | 35.9 (20.6) | 15.4 | 17.2 | 11.0 |
| PGIC NW | 5296 | 23.3 (4.8) | 19.2 | 27.6 | 25.1 |
| PGIC NO | 19571 | 16.6 (3.4) | 26.8 | 38.3 | 14.5 |
| PGIC NE | 49095 | 24.9 (23.0) | 20.1 | 19.5 | 12.6 |
| PGIC CE | 47129 | 34.5 (34.1) | 7.5 | 7.8 | 16.0 |
| PGIC SE | 24370 | 36.4 (33.9) | 8.5 | 5.0 | 16.2 |
| PGIC SW | 11421 | 36.5 (16.9) | 14.8 | 14.2 | 17.7 |
| TOTAL PGIC | 170904 | 29.9 (24.2) | 15.0 | 16.1 | 14.9 |
| TOTAL | 241813 | 28.4 (21.8) | 13.4 | 17.3 | 19.2 |

a buffer, also including lakes that do not directly intersect with the margin. As our ice mask has a higher resolution and hence granularity, the relatively similar absolute values result in a much smaller relative fraction of the total margin length, namely 3.3 % versus 10 % and 0.6 % versus 5 % for the GrIS and PGIC, respectively. This is still in the same order of magnitude as the length of marine termini, confirming the important role of lakes when investigating regional interactions of the ice with its surrounding terrain. This role that is expected to increase in future as marine termini become smaller and lake termini potentially larger (Ryan et al., 2024). However, the choice of underlying data and especially resolution of the highly complex margin, remains crucial when deriving regional estimates of the relative importance of the respective margin types.

The dominance of marine-terminating outlet glaciers in driving the mass balance of the GrIS is well documented and high flow velocities and high rates of discharge are generally concentrated around the relatively short fractions of the overall marine-terminating ice margin (Mouginot et al., 2019). However, our study highlights the importance of the land- and lake-terminating



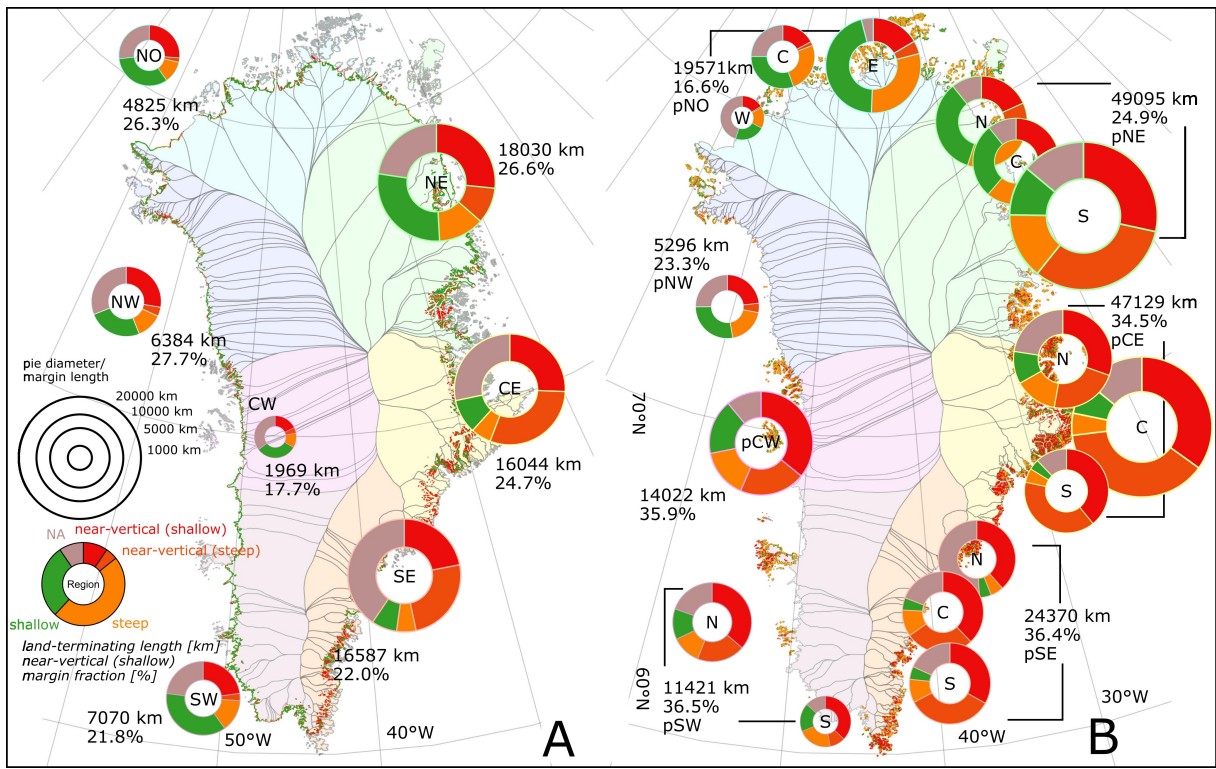

**Figure 9.** Distribution of margin morphologies along the land-terminating ice margin on the GrIS (A) and the PGIC (B). Morphologies include near-vertical margin over shallow as well as over steep terrain, steep margin sections as well as shallow ramps.

margin when considering the total area where ice interacts with surrounding terrain. The high total lengths for land-terminating
margin as well as their relative fractions of consistently above 93 % in the NE, CE and SE regions (both on the GrIS as well
as PGIC, Table 1) reflect the much more fractured and winding character of the margin in these regions. However, the land-terminating margin also dominates elsewhere and is above 85 % in all regions.

We rely on mosaics of the ArcticDEM product here, produced from imagery over multiple years, to assure a continuous cov-
erage and comparable quality of the tiles across Greenland. Relying on this data ignores changes along the margin that happen
within a few years. Due to the resolution of 2 m and the steep nature of large sections of the margin, the product is also not able
to definitely distinguish vertical sections from very steep sections of the margin. However, a comparison to higher resolution
Pléiades data suggests that the ArcticDEM is able to capture the margin morphology, with differences in slope less than half a
degree (Table 2). Chudley et al. (2021), relying on the same ArcticDEM mosaics to detect crevasses, have similarly noted the
product's quality to capture such features, with an arguably even more ephemeral nature. With the rapid development of high
resolution elevation data, especially with the inclusion of repeat IceSat and CryoSat data (Winstrup et al., 2024), investigations



of the margin, including its temporal change can be further refined.

Having identified approximately 25 % of the land-terminating margin as near-vertical on shallow terrain in NW, NO and NE
Greenland, respectively, suggests that the initial estimate of 45 % of the margin in North Greenland being dominated by ice
cliffs (Goldthwait, 1961), was exaggerated. Photographic evidence across North Greenland also suggests that likely much less
than half of the ice margin is actually ending in cliffs. It is possible that he understood this number to include cliffs as well
as very steep sections, which could not be accessed unaided, an initial interest of their research. The additional 11.6 to 12.6 %
found to be such ramps in the region, as well as 2.3 to 9.9 % of near-vertical margin over steep terrain are already much closer
to his initial proposition.

There is a large difference in morphology between the Western and Eastern margins of GrIS as well as PGIC (Table 3).
Why this is the case can not be conclusively answered here, but with closer scrutiny of the dataset it is obvious that many
near-vertical and steep sections in the eastern subbasins and along eastern PGICs are located along bedrock outcrops at higher
elevations. In North Greenland (with much smaller numbers in mountain and outlet glaciers) near-vertical sections appear
generally along straight sections of the ice sheet margin, as has been documented in field studies (Abermann et al., 2020; Kjær
et al., 2018), while in West and South Greenland they are more scattered along glacier lobes. The dominance of shallow ramps
in the far North (Peary Land) and the North-East (around the outlet of Zachariæ Isstrøm, Figure 9) correspond to the abundance
of relatively shallow ice caps (e.g. Hans Tausen Ice Cap, (Zekollari et al., 2017)) and vast expanses of land-terminating margins
in generally shallow terrain.

Near-vertical margin sections over steep terrain were on average located at highest elevations across all 14 regions ($\mu$=1129
m a.s.l., $\sigma$=402 m a.s.l.), with near-vertical sections over shallow terrain (870 m, 301 m), steep (750 m, 228 m) and shallow
margins (695 m, 198 m) following. This provides a first indication that margin morphology is not randomly distributed and
follows patterns that can be associated to either ice dynamics or climate. However, the high standard deviations point to the
strong variability between regions, which requires further scrutiny. With the present data, we are also able to show that with
the exception of Center-East and South-East Greenland, where lake-terminating margins are especially limited, near-vertical
margins over shallow terrain are more frequently directly adjacent to mapped lakes than other parts of the margin. 2.9 % of
near-vertical margin on shallow terrain directly connects to lakes present in the inventory, while only 2.1 % of the rest of the
margin do so. This points towards one of the original hypothesis, that ice cliffs are potentially formed when water bodies are
present next to the margin. However, to further investigate this, lake inventories from multiple years before the acquisition of
the DEMs and a more detailed analysis of margins with ice-lake interfaces are required.

Relying on available distributed data on bed topography (Morlighem et al., 2017), surface mass balance (Mankoff et al.,
2021), ice velocity (Solgaard et al., 2021) as well as climate (Noël et al., 2018) potentially allows to investigate which variables
determine the margin morphology, or inversely how the margin morphology potentially affects local ice dynamics and mass




change. As we believe that margin morphology is changing over time, with vertical sections having turned into ramps over recent decades (Abermann et al., 2020), future work should also investigate the temporal evolution of steep sections across the margin, to elucidate the potential of margin morphology informing about the state of the ice sheet and ice caps and their

potential climatic drivers. Previous research suggested that 1-10 % of local mass loss of a glacier in Antarctica can be attributed to dry calving when vertical margin is present (Fountain et al., 2006). Considering the large steep fraction of the margin in some subbasins on the GrIS as well as PGIC, a reassessment of the role of dry calving in local mass loss is desirable.

Our study allows for an interpretation of morphologies along the land-terminating margin. Where the margin intersects lakes,
confidence in the correct position of the ice masks decreases complex interactions and hence more dynamic conditions at the ice-water interface. We therefore refrain from a regional interpretation of margin morphologies along lake termini without manually delineating actual margin sections, as has been done before for marine termini Kochtitzky and Copland (2022). Future work should therefore consider mapping the more than 3000 km of lake-terminating margin across Greenland, including its temporal evolution at least a decade before the time stamp of ArtcicDEMs. This would allow to determine how ice mor-
phologies relate to the presence of lakes, or past occurrence of water bodies adjacent to the margin.

## 6 Conclusions

In this study we have provided a first comprehensive quantification of the relative length of the land-, marine- and lake-terminating margin in Greenland based on existing ice masks. We show that regionally between 85 % and 99 % of the ice
margin is land-terminating, with values ranging from 24 % to 100 % considering individual catchments. Although ice flux along the land-terminating margin is smaller compared to the equivalent margin length ending in water, this large total length of 241537 km of terrestrial ice margin underlines the importance of understanding associated processes, both for ice dynamics as well as the role the margin plays for peripheral hydrology, ecology and geomorphology. At the same time we also show that on the GrIS, lake-terminating margins cover relatively similar stretches (3.3 % of a total of more than 76000 km) as marine-
terminating margins (3.6 %) while values are smaller for PGICs (0.6% vs 1.6% of a total of more than 174000 km), confirming the overall important role lakes play when considering marginal ice dynamics in Greenland.

Our study shows the potential of the ArcticDEM products to investigate complex morphological features. The median difference in slope across an area of 8.5 km$^2$ along the margin at two study sites where a Pléiades DEM is available ranges only
between 0.2 and 0.5°. While the ArcticDEM is not able to capture vertical sections as perfect 90° planes, the clear distinction in slope distribution between shallow ramps, steep ramps and actual vertical sections is clearly visible and suggests the DEMs to be suitable to detect regional morphological patterns.



Extracted margin morphologies suggest that between 17.7 to 36.5 % of the land-terminating margin end in near-vertical sections, including vertical ice cliffs, over relatively shallow terrain in individual regions (and up to 50 % in some subbasins, especially frequent along the Northern GrIS margin and along PGICs in East Greenland). In general, more than half of the margin is composed of cliffs or steep ramps (>25°). More than half a century after Weidick noted the need to investigate the presence of ice cliffs at the regional scale in Greenland, we can show that this is now possible with relatively novel high resolution data. In contrast, shallow ramps are especially abundant along the North-Eastern and South-Western GrIS margin and on PGICs in North Greenland. Spatial variability is large and potentially driven by varying climate and underlying topography as well as upstream ice dynamics. The dataset produced in this study provides a basis for further investigations on the link between margin morphology, ice dynamics, bed topography and climate and can help to elucidate past and potential future patterns of the response of the ice margin at the interface with the proglacial terrain across Greenland.

*Code and data availability.* All data generated and associated code for this study is available at Steiner et al. (2025). Future development of the data is available at https://github.com/fidelsteiner/tIM.

Additional description of data is available in the Supplementary Material associated to this manuscript.

*Author contributions.* All authors conceived the study. JS performed the analysis and wrote the manuscript, with imput from JA and RP.

*Competing interests.* The authors declare no competing interests.

*Acknowledgements.* This research was funded in whole by the Austrian Science Fund (FWF) [grant DOI 10.55776/P36306]. The authors also acknowledge the Government of Greenland for support for an initial reconnaissance expedition in 2017 through the Tipps og Lottomidler Pulje C, that provided the basis for this study. The Pléiades stereo-pairs used in this study was provided by the Pléiades Glacier Observatory initiative of the French Space Agency (CNES). Greenlandic names were used where available, while the use of 'Greenland' was retained when referring to standard terms like 'Greenland Ice Sheet' or regional descriptions following previous studies.



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
