# Peer review of "The terrestrial ice margin morphology in Kalaallit Nunaat (Greenland)"

_EGUsphere, 2025_

## Author Comment (AC1)

**Response to comments on "The terrestrial ice margin morphology in Kalaallit Nunaat (Greenland)" (essd-2025-2424) by Steiner et al.**

Response to RC1 (Erin Pettit):

Review of Steiner et al

The terrestrial ice margin morphology in Kalaallit Nunaat (Greenland)

The authors state two goals. First, to provide the lengths of margins of the three main categories (marine, land, and lake terminating). And a second goal to show the variable morphology of the land-terminating glaciers, which range of shallow ramps to steep cliffs.

First - I really like that this paper is tackling the question of land-terminating glacier terminus morphologies. There is lots to be learned from the terminus behavior. So I really appreciate the authors efforts!

This paper overall describes the data set, and does not provide any scientific analysis alongside that data set. I admit that I expected - based on the title - to see some interpretation of the data. The last sentence of the abstract informs reader that this paper describes the data set and they leave the interpretation to others.

What I see as unique about this paper is the focus on the varying morphologies of the land-terminating ice margin. This goal, however, seems to take lesser importance in the abstract to the overall lengths of the margins. I would suggest that the authors emphasize this more novel data set and less on the cumulative distance that is marine/lake/land.

We appreciate the constructive reading of the manuscript. We agree that a large part of the study is circling around the production of a dataset. Considering that this hasn't been done before for the margin morphology and the nature of the land termini has previously not been described at this scale, we still consider this an analysis of data. However, what you characterize as the first goal has indeed taken too much center stage in our presentation of the results. While it consumed quite some work to accurately identify where the land terminating margin is located, it was simply a means to an end. Considering that compared to marine and even lake termini the land terminating margin has so far received relatively little attention in the literature on Greenland cryosphere, we felt it was important to state its relative magnitude. We also do not want the calculation of margin length (which, as we note, is purely based on the already existing margin products made by others) to be seen as the main take away, which has also confused others. While the numbers are still important, we now considerably reduce this part of the study and instead emphasize the aspect of morphology and how it was obtained further.

First the general categorization goal: While the categorizing of termini and determining total lengths of each can be important for some projects, without context for the numbers provided in the abstract, this is not so useful. Are those numbers bigger or smaller than expected? Are they changing? What year do they represent? I appreciate that they include all the margins, not just those with significant ice discharge - as the lateral margins of glaciers can play an important role. I looked to one tidewater glacier as an example in the data set - one where I have an upcoming project (69.79521923, -50.23659187) and saw 3.5km section of the northwest lateral margin (presumably against a fjord wall) flipping between land and tidewater, of a tidewater glacier as a mix of land and tidewater termini. Similarly the southeast margin has a section identified as tidewater that is not. This makes me question the overall quality control and the value of

publishing these numbers. The authors state in line 123 that they did visual quality control, so I am surprised that one random glacier picked seems to have errors - as does the glacier to the north. The data set is useful! - although specific use might require fixing the errors. For example, ice flux could be more accurately determined for different regions by the dot product of the normal to the margin and the ice flow direction across the margin - if both data sets are of sufficient resolution.

Thank you for checking the margin product itself – we share the same frustration when finding inaccurate margin classifications when identifying induvial areas but believe that this is an outcome of a pragmatic choice that also has advantages. We have considered whether we should manually classify the margin between marine, lake and land termini. While this would have likely reduced the number of misclassifications, it would also have resulted in a much more subjective dataset from region to region, without the chance to eventually reproduce the same approach as datasets like the ice mask are updated. Making the distinction purely based on elevation allowed us to assure a consistent quality across Greenland. As you note, the problem is especially accentuated along outlet/tidewater glaciers, where we are in transition from below to above sea level at the DEM. Also in response to the CC, we now account for this shortcoming by mapping a subsect of such locations, manually determining which parts are termini and which sections are at the sides of outlet glaciers, with a DEM < 10 m a.s.l. but obviously no direct ice-water interface. We suggest that this provides a possibility to interpret the absolute number of what we call 'marine-terminating' margin with respect to their useability. We also would like to emphasize again, and have done so now more clearly in the Methods as well as the Results, that our marine margins should not be seen as termini with actual ice discharge. For that – like you suggest – a point location of the actual terminus location would be useful (and ideally placed by the user, depending on what application is attempted) or datasets from other studies (Kochtitzky et al., 2022; Kochtitzky & Copland, 2022; Ryan et al., 2024) is more appropriate.

My suggestion is that this primary margin categorization be less of the end goal, and less emphasized in the abstract - unless the authors want to interpret their results in context of the Ryan et al 2023 and other papers that have done similar work using a slightly different approach and goals. And please be more clear that quality control is weak (i.e. line 123) - so the data should be used with caution.

We agree that the margin characterization is not the primary goal here, especially with respect to the non land-terminating sections. We have now also made the description of quality control clearer and note that it should be used with caution, which we believe is reasonable with a product at this scale. With the discussion of other potential ice masks (see response to RC2), this section has now also considerably expanded.

Towards their goal of the shape of the land terminus - this I find is a unique and useful contribution, as the shape of the terminus reflects the interaction of physical processes (ice flow with fracturing and details of surface mass balance). And I believe we can learn about evolving ice dynamics from these margins.

We agree that it would be great to have a much more in depth analysis with a number of other – ice dynamic and possibly also climatic – variables and their potential role in explaining variables. The work we do here we would hope is a segway to be able to eventually do that. Since, however, at this resolution getting such data isn't trivial without having to explain also their intrinsic challenges we felt that with the space we already needed to reasonably explain how we derived the margin morphologies, this would have gone well beyond a manuscript where we can do these

aspects justice. For ice flow for example we would need to determine the direction at the respective – not always very straight – margin sections, climate data is often not available in appropriate resolution and linkages to morphology that has evolved over decades, centuries or longer are not trivial. We have now made substantial changes to the analysis focusing specifically much more in the morphology and less on the margin outline itself and hope that this allows for a solid basis to do this kind of analysis (as well as the analysis of how this morphology is changing in time) in subsequent steps. We now also emphasize these potential future steps more clearly in the Discussion.

The paper describes the methods for delineating steep versus shallow ramps in a clear way - I am not an expert in remote sensing analysis of this kind, so I don't feel like I can speak to those details, but it generally seems robust except for the issue of it being clear what dates at attached to the results, so they can be used to look at changes.

We agree that this hasn't been clarified clearly (also in metadata) and we have now amended this. Following changes from the response to RC2 there have been substantial changes regarding dealing with timing of the products, which resulted in substantial changes in this general description, but also now includes a clear communication of time frame the product is valid for.

If the authors choose not to do any analysis with their data set, I would suggest one additional statistical result might be helpful for understanding the data - perhaps for quality control, perhaps for science. That is to look at the distribution of lengths of margin segments of each type (separately doing the land/lake/marine versus the steep/shallow). For the general margin categories, this kind of statistical analysis might point to errors - short sections of marine terminating margin <500m that are separated from a longer section of marine terminating might point to places where the categorization is in error (such as the location I identified above).

Thank you for the suggestion. We agree this is a worthwhile undertaking and now provide this analysis before the discussion of the morphology and replacing some of the earlier text that only presented raw numbers on margin lengths.

For the steep/shallow categorization on the other hand, the distribution of many short segments versus fewer longer segments might offer some insight into the dynamics.

Thanks for the suggestion. Considering we have now reduced the discussion of results from simple margin statistics, we now include a more in depth discussion of such patterns (in the Discussion section), as far as that is possible with additional discussion of variables like ice dynamics, an analysis that is definitely of interest but we think goes beyond the scope of a manuscript like this one invested in the establishment of the morphology dataset.

**General comments about the structure:**

The introduction spends a lot of time motivating why we want to know the margins - especially the land terminating margins. That approach seems good, but the discussion and conclusions don't really come back to these ideas - instead the discussion focuses on lengths of different categories and the value of Arctic DEM. The discussion/conclusions that relate back to the introductory material is very general - mostly saying this is future work. My suggestion is better balancing the physical explanations introduced early in the paper with the discussion/conclusions so that there feels like some closure for the reader. If the authors want to emphasize that the biggest gap in answering physical questions is the lack of good data set - then

emphasize that more in the introduction, why has there been a lack of sufficiently good data sets to tackle the question of steep versus shallow? Is it just the morphology data set? What other data sets do we need to answer some of these questions? While they suggest this data set is the primary one lacking in line 80, they could expand on this more - and clearly lay out the gap in knowledge they are trying to fill. To summarize my thinking here - the introduction suggests that the authors are going to assess the physical processes more than they do. The discussion/conclusion don't really say much except that their data set might be helpful for future work (without many details).

We are very grateful for the close reading on the general scope. We actually thought that our main aim always was the morphology and the retrieval from the DEM products but agree that in the course of writing it this hasn't come through well enough the way things were structured and emphasized. We have now significantly reduced the aspects of margin outline analysis (adding a few suggestions that have been made to also further sharpen this and revised the product following comments from all reviewers), provide a clearer discussion on the reasoning why this data is important and where there was a gap as well as providing clearer framing on what future analysis, beyond what is reasonable in the space here, is feasible, including what data is needed, where this is at all useful and realistic and where the focus of such analysis can lie (also considering the quality and shortcomings of the present dataset).

I felt like the paper was overall well written and explained their method clearly, I do not have many suggestions details of wording in these areas.

We are very grateful for the close reading on methods and comments.

I am happy to chat more with the authors if they wish.

**References:**

Kochtitzky, W., & Copland, L. (2022). Retreat of Northern Hemisphere Marine-Terminating Glaciers, 2000–2020. *Geophysical Research Letters*, 49(3), e2021GL096501. https://doi.org/10.1029/2021GL096501

Kochtitzky, W., Copland, L., Van Wychen, W., Hugonnet, R., Hock, R., Dowdeswell, J. A.,

Benham, T., Strozzi, T., Glazovsky, A., Lavrentiev, I., Rounce, D. R., Millan, R., Cook, A.,

Dalton, A., Jiskoot, H., Cooley, J., Jania, J., & Navarro, F. (2022). The unquantified mass loss of Northern Hemisphere marine-terminating glaciers from 2000–2020. *Nature Communications*, *13*(1), 5835. https://doi.org/10.1038/s41467-022-33231-x

Ryan, J., Ross, T., Cooley, S., Fahrner, D., Abib, N., Benson, V., & Sutherland, D. (2024). Retreat of the Greenland Ice Sheet leads to divergent patterns of reconfiguration at its freshwater and tidewater margins. *Journal of Glaciology*, 1–9. https://doi.org/10.1017/jog.2024.61

---

## Author Comment (AC2)

**Response to comments on "The terrestrial ice margin morphology in Kalaallit Nunaat (Greenland)" (essd-2025-2424) by Steiner et al.**

Response to RC2 (Anders Björk):

The Steiner et al team deliver the first Greenland-wide quantification of land-, lake-, and marine-terminating margins and a spatial census of terrestrial margin morphologies for both GrIS and Greenlandic PGICs. This is a major accomplishment, and a very usfull scientific contibution for numerous future process and change detection studies. The paper's immmediate reproducibility makes it a foundational dataset and method for ice margin research in general.

This manuscript is clear, well-structured, and methodologically careful. The authors combine established datasets with sensible preprocessing, and a distribution-based slope classification, validated against Pléiades DEMs. Uncertainties and limitations are quantified and transparently handled, and all code/data are made available.

I really applaud the authors for doing this work and for taking the time to develop the method. I am confident, that it will be a method and procedure, which will be used on many datasets in the future, and provide critical knowledge and understanding of our ice margins and their changes.

Thank you very much for the constructive reading and encouraging response. We address all points raised comment by comment below.

I have one major point, which requires attention:

My major point of concern with the study in its current form is the use of the PROMICE ice mask (Citterio & Ahlstrøm, 2013). This particular outline represents a manually derived ice margin based on aerial photographs collected in the period 1978-1987, and not a year-2000 margin as the authors interpret. This discrepancy results in a potential large glacial retreat between the timing of the margin and the timing of the DEM used for the analysis. The authors do initiate a series of measures to counter the supposed offset from 2000-2012, but the actual off set in timing is much larger.

My concern is that too many cells have been excluded as a result of this, and only regions where retreat since the mid-1980s have been minimal are included. Under all circumstances, a 100 meter buffer from the 1980s margin, will many regions be inadequate as reported retreat rates are often in the order 5-20 meters / year. This concern is illustrated by figure 4d, where some of the excluded 1km grid cells, show a frontal retreat (between ice mask and DEM) of more than 500 m. Here most of the margin is excluded, and as a result, knowledge of ice margin slope of the rapid retreating ice margins are omitted.

There are a number of ways to go about this, but unfortunately I don't see any that does not requires substantial extra work. There are newer datasets available that offer an ice margin, closer in time to the ArcticDEM. One option is "OpenLand" from the Danish Climate-data Agency. This outline is from 2017-21 https://dataforsyningen.dk/data/4771. Another option is a beta-dataset from GEUS – the new PROMICE-2022 ice mask, which is currently under review, but with data available https://essd.copernicus.org/preprints/essd-2025-415/, however this is only covering the ice sheet and not the PGICs. This could however be combined with one of the newer PGIC outlines like the Randolph Glacier Inventory which match closer in time to the ArcticDEM.

If the authors argue that the 1980s outline is sufficient, based on the extensive three step approach with visual inspections, I would like to see a more comprehensive analysis of the effects of the excluded ice margins: How much of the margin is visually inspected? What are the cut-off values for a cell to be discarded? What would be the effect of a different buffer size?

I would expect the excluded cells to be in the lower elevation parts of the terminating margins (the glacier front), which will skew the results more towards lateral margins, with potential substantial implications for the overall results and conclusions.

Thank you for pointing out the serious issue on the mask timing. Even more embarrassingly, the error made was here was going two ways. While we have initially used the PROMICE mask from the 1980s (as we were working with older Korsgaard DEMs, where the temporal overlap works well), the final mask we employed was the CCI product, that was published together with the product for PGICs (Rastner et al., 2012) and hence – with all its deficiencies – is also of comparable quality. We regret this lapse in the documentation of the data used. The advantage of the CCI mask is that it does align closer to the time step of the DEMs (having been produced from imagery between 1994 and 2004). This doesn't absolve us however of (a) the general issue of quality, (b) the mismatch especially around the termini of outlet glaciers (also those terminating on land) and (b) remaining temporal offset to the DEMs and the adequacy of the buffer. We appreciate your suggestion of the additional datasets, which we have now also considered to make the analysis more robust. Considering these aspects we would like to outline below how we address the issues raised more concretely:

a) We understand the concern that a relatively small buffer removes many potentially interesting pixels, but we also found that an increased buffer (200 m, 500 m) makes the detection of the near-vertical sections increasingly difficult as they are often very short, result in then very few pixels and in the approach we chose are overshadowed by many shallow pixels on the ice. We now provide a sensitivity analysis and show this in a separate section of the methodology for well known sections of the margin. This naturally comes at a trade-off and is especially problematic when the margin isn't placed accurately and no ice at all is captured in the buffer. For this we have shown on 41 locations across Greenland (Supplementary Material) that this buffer works well. However, considering the restructuring of the manuscript, reducing the parts on the margin lengths and emphasizing the aspect of mapping the morphology, we have now expanded this both in scope and scrutiny, and place the discussion into the main manuscript, to have a more solid discussion of the uncertainty of the product. We now place 200 such profiles across the margin, randomly placed and extract the morphology. From this we determine the true positives, where the margin was obviously captured. We would also like to highlight that we do provide grid cells where the margin is obviously so misplaced that we do not calculate the morphology. We furthermore now emphasize in the manuscript that the margin morphology at outlet glacier termini needs to be handled with caution, due to its rapidly changing nature (see for example Figure 1). This introduces to some degree skewness you mention. We address this for the overall product by showing for each basin, where in the elevational distributions the mismatch is larger. This reduces the number of locations where we extract the morphology, with the current quality of the masks and us being reluctant to manually adjust it (which would hamper reproducibility in equal quality) we believe this is acceptable. We now also provide a more detailed discussion of the patterns of morphology across Greenland at the scale beyond small sections.

- b) We now also quantify the amount of margin that was visually inspected and deemed insufficient and provide a quantitative measure when this was done so (following the mask and imagery, with an obvious mismatch beyond 100 m).
- c) As noted, the outline we used does match the ArcticDEM more closely than the PROMICE outline from the 1980s, hence we believe the buffer chosen in general is acceptable. However, we agree that there are many cases when there is a mismatch. The suggestion of the alternative margin products has now prompted us to evaluate the differences and discuss the pros and cons of using one or the other, which like you also note, is in any case of essence as potential future iterations will definitely need to be carried out with a new margin mask. We believe that checking individual sites in such a large product is always somewhat problematic as a definite quality check and the perfect match is unlikely possible. Figures 1 to 3 show a couple of examples, where we superimpose all masks against the imagery from 2000. The SDFE data (the dataset you kindly suggested, i.e. from 2017-2021) seems to have excellent quality at many margins and would undoubtedly be the best temporal fit for the ArcticDEM. Unfortunately, it has also many missing sections and some glaciers seem to have been mapped rather randomly (see Figure 1 and Figure 3). Since its documentation is not available like for any of the other masks, we feel it may be unwise to use it for a product where future iterations are foreseen. The PROMICE 2022 dataset, still in review, is likely the currently best option for the ice sheet, but we are faced with the mismatch in time and the possibility that parts of the mask are within the actual ice sheet on the ArcticDEM and we lack the PGIC data. The CCI data obviously has many erroneous parts, but given the location with the buffering, congruent with the product we use for PGICs is in our eyes the most pragmatic and consistent choice. However, to quantify the sensitivity of this choice, we now extract morphologies with the different margins and show the ability to capture the margin on the DEM for sections of relative stability across all decades, as well as for lower regions of outlet glaciers that show more rapid changes.

Figure 1: (left) The four available masks on a glacier in Western Greenland where a co-author carried out field research (imagery 2000). Note that the SDFE data is accurate at the lower margin, but only available for the tongue and no adjacent areas. (right) Same glacier with manual delineations, imagery (1980s)

Figure 2: Margin section in central Western Greenland, showing sections of relatively close alignment for all margins, also suggesting relative stability. Imagery from 2000, which should match the CCI margin.

Figure 3: (left) Margin in western Greenland, where for some reason the CCI margin is especially off. (right) Margin in Eastern Greenland, where both the CCI but also the newest PROMICE 2022 product show artefacts where there is clearly no icein the top center part, relatively good match for all products on the glacier tongue except for the lowest terminus and mismatch for the CCI product on the ice cap on the right. Note the SDFE margin only being available for some parts of the ice cap.

Many scientists would want such a study to also include change over time, and given the multitude of datasets available, it is indeed also possible, eg using the AeroDEM (Korsgaard et al, 2016) which corresponds exactly in time to the 1980s PROMICE-ice mask. However, I don't see it as a prerequisite for publishing this study. The work in itself, and the dataset, is sufficient to warren a major scientific contribution, and I am confident that several later change-studies will develop from this paper.

We absolutely agree with this sentiment but felt that for the amount of further methodology, this would have gone beyond the reasonable space here. Noteably, it would require to treat the ArcticDEM product somewhat differently. We would need to take apart the DEMs according to their timesteps, which luckily is possible with the ArcticDEM dataset. However that would leave

us with only a subset of regions for the specific times (late 2000s/early 2010s vs early 2020s) as all of Greenland is not covered every time. It would then likely also require to use two different margins for these distinct steps, especially for faster changing regions. We believe that this is where now the new PROMICE 2022 becomes interesting. In response to reviewer 2, we have now also expanded the section on the analysis steps that can be taken with this product further down the line, and now also include this aspect there in more detail.

---

## Author Comment (AC3)

**Response to comments on "The terrestrial ice margin morphology in Kalaallit Nunaat (Greenland)" (essd-2025-2424) by Steiner et al.**

**Response to CC1 (Jonathan Ryan):**

We are grateful for this community comment and your careful checking of the data and respond to all points below.

I just wanted to alert the authors that Ryan et al. (2024) mapped lengths of marine- and lake-terminating (and land-terminating by subtraction) margins for the Greenland Ice Sheet (see citation below). Some of the text in your manuscript therefore slightly overstates its significance (e.g. L4 and L358: "for the first time..." and "first comprehensive quantification..."). I think the statements at L4 and L358 should be revised to acknowledge this.

We regret having missed including the findings you have already made in Ryan et al. (2024). While we have included this publication in our discussion of the results, we have indeed missed its acknowledgement in our discussion of margin lengths. In response to reviewer suggestions, our framing of the results with respect to these numbers has also changed (moving somewhat away from the emphasis on lengths, which was indeed not the study focus), hence the text in these sections has changed overall, but we naturally now also compare to the findings from your study.

I was not able to reproduce the total values in Table 1 for the GrIS. I found that the total length of *Regional\_Lake\_Margin\_GrIS.gpkg* is 6,446 km which would be 8.5% of the total perimeter. Likewise, *Regional\_Marine\_Margin\_GrIS.gpkg* has a total length of 12,138 km which would be 16.0% of the perimeter. Maybe I did something wrong – I've included my code in the attached PDF.

We apologize for the error in compiling the data when pulling it into gpkgs, where segments were duplicated. After the complete revision following the suggested changes from the review we will provide an updated dataset in the repository that naturally nees to match the data presented in the manuscript.

The length of GrIS margin is longer than Ryan et al. (2024) (76,154 vs. 29,269 km). I think the main reason for the differences is the treatment of nunataks which you include (but we exclude). It looks like you are able to provide statistics with and without nunataks. It would be great if you could provide two numbers (i.e. with nunataks included and excluded) throughout the manuscript so that we can more directly compare our findings.

The treatment of nunataks was indeed a source of concern – while we think they are probably less interesting in terms of understanding any kind of flux processes and to some degree even melt runoff, in the end they still constitute margin and may cease to be nunataks in future as the ice retreats. Hence, their inclusion also seems warranted. However, we agree that it's crucial to

have clarity on where they are included and where not. We have now throughout (specifically in tables) stated numbers for the analysis with and without nunataks.

The length of the GrIS ice-ocean boundary looks like it is overestimated (12,138 km for GrIS). It looks like the dataset incorrectly identifies some nunataks as ice-ocean boundaries. There are also many cases where the sides of tidewater glaciers are identified as ice-ocean boundaries. See attached PDF for a couple of examples. Note that Ryan et al. (2024) found the GrIS ice-ocean boundary to be 1,598 km in 1990-95 and 1,439 km in 2003-07. The large differences between the two numbers should at least be mentioned in the Discussion.

The big difference comes from a rather stringent definition we took here based on elevation from the Bedmachine product, which we did not manually verify except for completely erroneous regions due to glitches in the DEM. As a result, like you note, the sides of tidewater glaciers that do not have a water interface but with respect to the DEM are in a location where the sea level is higher or nearly the same, are considered as marine termini as well, even though here would be no flux, or no interaction between water and ice. To avoid this problem, we initially considered manually mapping all parts of the margin located at locations below sea level but not actual a marine termini, but realized that this wasn't trivial in many locations and would add a lot of subjectivity on an issue that wasn't the focus in this study. To however further emphasize this issue, we have now mapped this for a subset of tidewater glaciers where the distinction is clearly possible, calculate the fraction of clear termini and non-termini shear margins and other lateral sides that are marked marine-terminating by our approach. This allows for an estimate for a general fraction of these different sections across Greenland. We have now also more clearly emphasized that our marine termini should not be misconstrued termini with flux gates, in both Methods and Results.

There is also a large difference between the length of the GrIS ice-lake boundaries between this study and Ryan et al. (2024) (6,445 km vs. ~550 km). I understand that the ice-lake boundaries are more challenging to identify but, again, it would be useful to mention whether these differences are caused by decisions to include vs. exclude nunataks in the Discussion given the similar goal of both datasets.

For lakes we are faced with a more complex issue of other studies actually having identified even larger margins (>3000 km (Carrivick et al., 2022)) over our length of >2500 km (see Table 1), visualizing the large differences for different methods. We now also add your estimate in the Discussion and provide a clearer discussion on how these differences can be explained (nunataks in the comparison to your data being one case, the fact that we mapped additional lakes to the dataset by (How et al., 2021) where they were missing for the 2000 time step). We try to quantify as much as possible where these differences come from based on these considerations. For this we compare some of the termini regions from both studies and provide the comparison in the Supplementary Material.

Thanks and good luck with the rest of the review process.

**References**

- Ryan, J., Ross, T., Cooley, S., Fahrner, D., Abib, N., Benson, V., & Sutherland, D. (2024). Retreat of the Greenland Ice Sheet leads to divergent patterns of reconfiguration at its freshwater and tidewater margins. Journal of Glaciology, e65. https://doi.org/10.1017/jog.2024.61
- Carrivick, J. L., How, P., Lea, J. M., Sutherland, J. L., Grimes, M., Tweed, F. S., Cornford, S., Quincey, D. J., & Mallalieu, J. (2022). Ice-Marginal Proglacial Lakes Across Greenland:

  Present Status and a Possible Future. *Geophysical Research Letters*, 49(12), e2022GL099276. https://doi.org/10.1029/2022GL099276
- How, P., Messerli, A., Mätzler, E., Santoro, M., Wiesmann, A., Caduff, R., Langley, K., Bojesen,
  M. H., Paul, F., Kääb, A., & Carrivick, J. L. (2021). Greenland-wide inventory of ice
  marginal lakes using a multi-method approach. *Scientific Reports*, 11(1), 4481.
  https://doi.org/10.1038/s41598-021-83509-1